# Counteranion-induced structural isomerization of phosphine-protected $PdAu_8$ and $PtAu_8$ clusters

Yu Fujiki[1], Tomoki Matsuyama[1], Soichi Kikkawa[1,2], Jun Hirayama[1,2], Hikaru Takaya[3], Naoki Nakatani[1], Nobuhiro Yasuda[4], Kiyofumi Nitta[4], Yuichi Negishi [5] & Seiji Yamazoe [1,2,6 ✉]

Controlling the geometric structures of metal clusters through structural isomerization allows for tuning of their electronic state. In this study, we successfully synthesized butterfly-motif $[PdAu_8(PPh_3)_8]^{2+}$ (**PdAu8-B**, **B** means butterfly-motif) and $[PtAu_8(PPh_3)_8]^{2+}$ (**PtAu8-B**) by the structural isomerization from crown-motif $[PdAu_8(PPh_3)_8]^{2+}$ (**PdAu8-C**, **C** means crown-motif) and $[PtAu_8(PPh_3)_8]^{2+}$ (**PtAu8-C**), induced by association with anionic poly-oxometalate, $[Mo_6O_{19}]^{2-}$ (**Mo6**) respectively, whereas their structural isomerization was suppressed by the use of $[NO_3]^-$ and $[PMo_{12}O_{40}]^{3-}$ as counter anions. DR-UV-vis-NIR and XAFS analyses and density functional theory calculations revealed that the synthesized $[PdAu_8(PPh_3)_8][Mo_6O_{19}]$ (**PdAu8-Mo6**) and $[PtAu_8(PPh_3)_8][Mo_6O_{19}]$ (**PtAu8-Mo6**) had **PdAu8-B** and **PtAu8-B** respectively because **PdAu8-Mo6** and **PtAu8-Mo6** had bands in optical absorption at the longer wavelength region and different structural parameters characteristic of the butterfly-motif structure obtained by XAFS analysis. Single-crystal and powder X-ray diffraction analyses revealed that **PdAu8-B** and **PtAu8-B** were surrounded by six **Mo6** with rock salt-type packing, which stabilizes the semi-stable butterfly-motif structure to overcome high activation energy for structural isomerization.

[1] Department of Chemistry, Graduate School of Science, Tokyo Metropolitan University, 1-1 Minami-Osawa, Hachioji, Tokyo 192-0397, Japan. [2] Elements Strategy Initiative for Catalysts & Batteries (ESICB), Kyoto University, 1-30 Goryo—Ohara, Nishikyo—ku, Kyoto 615-8245, Japan. [3] Department of Life & Health Sciences, Teikyo University of Science, 2-2-1 Senjyusakuragi, Adachi-ku, Tokyo 120-0045, Japan. [4] Center for Synchrotron Radiation Research, Japan Synchrotron Radiation Research Institute (JASRI), 1-1-1 Kouto, Sayo-cho, Sayo-gun, Hyogo 679-5198, Japan. [5] Department of Applied Chemistry, Faculty of Science, Tokyo University of Science, 1-3 Kagurazaka, Shinjuku-ku, Tokyo 162-8601, Japan. [6] Precursory Research for Embryonic Science and Technology (PRESTO), Japan Science and Technology Agency (JST), Kawaguchi, Saitama 332-0012, Japan. ✉email: yamazoe@tmu.ac.jp

Metal clusters of less than 100 atoms have attracted substantial interest because they have quantized electronic structures and unique geometric structures that cannot be predicted from the bulk metals and metal nanoparticles[1]. Since their electronic structures crucially depend on the cluster sizes, compositions, and geometric structures at the atomic level, atomically precise synthesis is required to understand the chemical and physical properties of the clusters. Recent progress in precise synthesis techniques has enabled the synthesis of a variety of gold, silver, copper, and alloy clusters protected by organic ligands such as thiolate, phosphine, alkynyl, carbene, and polyoxometalate ligands, with atomic precision[2–9]. Their unique catalytic, optical, magnetic, and redox properties related to geometric and core-ligand interfacial structures have been reported.

Structural control of ligand-protected metal clusters is one of the key factors in tuning the electronic state of these clusters. Numerous structural isomers have been reported using the structural flexibility of the metal cluster core and core-ligand interface. The structural isomers of metal clusters were found in $[Au_9(PR_8)_9]^{3+}$[6], $[Au_9Ag_{12}(SR)_4(dppm)_6X_6]^{3+}$[10], and $[Au_4Cu_4(L)_7]^{+}$[11]. The protecting ligands with different steric structures gave us structural isomers with different core structures such as $[Au_{11}(PR)_n]^{3+}$[12], $[Au_{18}(SR)_{14}]$[13], $Au_{24}(L)_{20}$[14], $Au_{25}(SR)_{18}$[13], $Au_{30}(SR)_{18}$[13,15], $Au_{52}(SR)_{32}$[16], and $Au_{144}(SR)_{60}$[17]. The control of geometric structures by using external stimuli has been demonstrated. Ligand exchange induced the reversible isomerization of $Au_{28}$ clusters[18]. Thermal-induced isomerization has been reported for $Au_{38}(SR)_{24}$[19], $Pd_2Au_{36}(SR)_{24}$[20], and $Cu_{15}(C≡CR)_{10}(L)_5$[21]. The coupling/decoupling of cationic surfactants provided the reversible isomerization of $[Au_{25}(p\text{-}MBA)_{18}]^{-}$ and collective rotation of the $Au_{13}$ core in the cluster has been proposed in the conversion process[22]. Structural isomerization by electrochemical redox has been found in the phosphine-protected $Au_8$ cluster[23]. Using high-resolution transmission electron microscopy, it was observed that the core structure of $Au_{144}(SR)_{60}$ was isomerized between icosahedral and face-centered cubic structures[24]. The $[Au_9(PR)_8]^{3+}$, $[PdAu_8(PR)_8]^{2+}$, and $[Au_{25}(SR)_{18}]^{-}$ have been reported to be isomerized by gas phase collision[25,26]. The structural isomerization from unstable $Au_{38}$ to stable $Au_{38}$ clusters was induced by heating, and their optical absorption, fluorescence properties, and catalytic activities depended on their structures[27].

It was previously reported that the core structure of $[Au_9(PPh_3)_8]^{3+}$ could be controlled by counteranion[28–32]. The crown motif $[Au_9(PPh_3)_8]^{3+}$ (**Au9-C**) in solution was maintained by the formation of salt with $[PMo_{12}O_{40}]^{3-}$, whereas the butterfly-motif $[Au_9(PPh_3)_8]^{3+}$ (**Au9-B**) was formed by the association with $[NO_3]^{-}$ or $Cl^{-}$ as counteranions because of the soft Au–Au bonds in the $Au_9$ core[32]. The structural isomerization of $[Au_9(PPh_3)_8]^{3+}$ was also induced by pressure[33] and gas phase collision[26]. The central Au atom of **Au9-C** can be substituted by Pd and Pt to synthesize bimetallic $[PdAu_8(PPh_3)_8]^{2+}$ (**PdAu8-C**) and $[PtAu_8(PPh_3)_8]^{2+}$ (**PtAu8-C**) with a crown-motif structure. In the cases of **PdAu8-C** and **PtAu8-C**, structural isomerization has yet to be reported despite them having the same geometric structure as **Au9-C**. One possible reason for the suppression of structural isomerization is the formation of stiff metal bonds by hetero metal atom doping[32]. Recently, it has been proposed that the isomerization of **PdAu8-C** occurs via gas phase collision[26]. This interesting possibility prompted us to hypothesize that the isomerization of **PdAu8-C** and **PtAu8-C** can be induced by applying strong external stimuli.

The cationic $[Au_9(PPh_3)_8]^{3+}$ and $[MAu_8(PPh_3)_8]^{2+}$ (M = Pd and Pt) associate with anionic compounds such as $[NO_3]^{-}$, $Cl^{-}$, and polyoxometalates to form ionic solids by coulomb interaction. In the present study, we synthesized the salts of $[MAu_8(PPh_3)_8]^{2+}$ using $[NO_3]^{-}$, $[PMo_{12}O_{40}]^{3-}$ and $[Mo_6O_{19}]^{2-}$

as counteranions with different sizes and charges. We successfully synthesized isomers from the crown motif of **PdAu8-C** and **PtAu8-C** by using $[Mo_6O_{19}]^{2-}$. The isomers showed different optical properties from **PdAu8-C** and **PtAu8-C** and had an absorption band in the longer-wavelength region. X-ray absorption fine structure (XAFS) analysis, which is a powerful tool to determine the local structure (coordination number, bonding distance) for each element, and density functional theory (DFT) calculations revealed that the isomers were butterfly-motif $[PdAu_8(PPh_3)_8]^{2+}$ (**PdAu8-B**) and $[PtAu_8(PPh_3)_8]^{2+}$ (**PtAu8-B**). Here, we also discuss the structural isomerization of **PdAu8-C** and **PtAu8-C** from the perspective of the activation energy for isomerization and crystal packing of clusters.

## Results

**Optical properties of PdAu8-Mo6 and PtAu8-Mo6.** The synthesis of $[Au_9(PPh_3)_8](NO_3)_3$ (**Au9-NO3**), $[PdAu_8(PPh_3)_8](NO_3)_2$ (**PdAu8-NO3**), and $[PtAu_8(PPh_3)_8](NO_3)_2$ (**PtAu8-NO3**) was confirmed by UV-Vis-NIR spectroscopy in ethanol solution, ESI-MS, and $^{31}P$-NMR, as shown in Fig. 1 and Supplementary Figs. S1–S3. The UV-Vis-NIR spectra of **Au9-NO3**, **PdAu8-NO3**, and **PtAu8-NO3** in ethanol solution in Fig. 1 and Supplementary Fig. S1 showed the characteristic optical properties of crown-motif **Au9-C**, **PdAu8-C** and **PtAu8-C**, respectively[34]. ESI-MS and NMR spectra in Fig. 1 and Supplementary Figs. S2 and S3 of **Au9-NO3**, **PdAu8-NO3**, and **PtAu8-NO3** indicated the presence of $[Au_9(PPh_3)]^{3+}$, $[PdAu_8(PPh_3)]^{2+}$, and $[PtAu_8(PPh_3)]^{2+}$ without other species, respectively. In addition, we confirmed that Pd and Pt in **PdAu8-NO3** and **PtAu8-NO3** were located at the center of crown-motif structures by Pd K- and Pt $L_3$-edges FT-EXAFS analysis as shown in Supplementary Fig. S9 and Supplementary Table S1 (discussed in "Local structures of PdAu8-Mo6 and PtAu8-Mo6"). ESI-MS and TG-DTA spectra of $TBA_2$ $[Mo_6O_{19}]$ (**TBA-Mo6**, TBA: tetrabutyl ammonium) suggested the successful synthesis of Lindqvist-type $[Mo_6O_{19}]^{2-}$ (Supplementary Figs. S4 and 5).

Figure 1C shows UV-Vis-NIR spectra of composites of $[PdAu_8(PPh_3)_8]^{2+}$ and cations. The DR-UV-Vis-NIR spectra of **PdAu8-NO3** and $[PdAu_8(PPh_3)_8][HPMo_{12}O_{40}]$ (**PdAu8-PMo12**) powders resembled that of **PdAu8-NO3** in ethanol solution because both **PdAu8-NO3** and **PdAu8-PMo12** have been reported to have a crown-motif **PdAu8-C** structure[32,34,35]. Optical properties of $[PdAu_8(PPh_3)_8][Mo_6O_{19}]$ (**PdAu8-Mo6**) differed from those of **PdAu8-NO3** and **PdAu8-PMo12**, and an absorption peak appeared at a longer-wavelength region (703 nm). Similar change in the optical property was observed in the case of $[PtAu_8(PPh_3)_8]^{2+}$, as shown in Fig. 1D. The DR-UV-Vis-NIR spectrum of $[PtAu_8(PPh_3)_8][Mo_6O_{19}]$ (**PtAu8-Mo6**) had broad absorption at 600–700 nm whereas **PtAu8-NO3** and $[PdAu_8(PPh_3)_8][HPMo_{12}O_{40}]$ (**PtAu8-PMo12**) with a crown-motif structure showed optical absorption similar to that of crown-motif **PtAu8-NO3** in ethanol solution. In the case of non-doped $[Au_9(PPh_3)_8]^{3+}$, butterfly-motif **Au9-B** was formed in **Au9-NO3** composite, which was formed from **Au9-C** ethanol solution by isomerization[32]. DR-UV-Vis-NIR spectrum of **Au9-NO3** with a butterfly-motif structure showed a characteristic absorption peak at 687 nm, which differed from those of **Au9-C** solution (**Au9-NO3** in ethanol) and crown-motif $[Au_9(PPh_3)_8]$ $[PMo_{12}O_{40}]$ (**Au9-PMo12**)[34]. Since the optical property of $[Au_9(PPh_3)_8][Mo_6O_{19}]$ composite (**Au9-Mo6**) was in accordance with that of **Au9-NO3** composite as shown in Supplementary Fig. S1, the **Au9-C** in **Au9-NO3** ethanol solution was isomerized to **Au9-B** by the association with **Mo6**. The formation of **Au9-B** isomer and the appearance of an absorption peak in the longer-wavelength region in **Au9-Mo6** gave us the idea that the

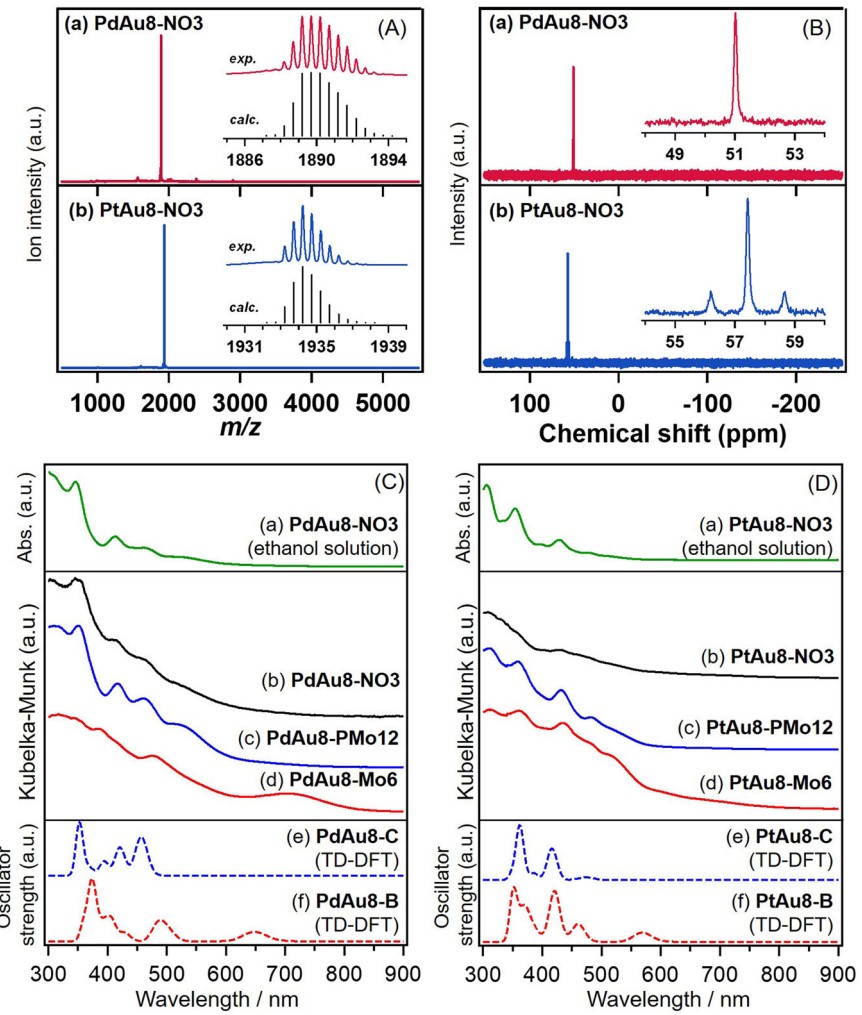

**Fig. 1 Mass analysis and optical properties. A** Positive-ion ESI mass spectra of (a) **PdAu8-NO3** and (b) **PtAu8-NO3** acetonitrile solution with experimental and calculated isotope patterns. **B** [31]P NMR spectra of (a) **PdAu8-NO3** and (b) **PtAu8-NO3** in chloroform-d. **C** UV-Vis-NIR spectra of (a) **PdAu8-NO3** in ethanol solution, DR-UV-Vis-NIR spectra of (b) **PdAu8-NO3**, (c) **PdAu8-PMo12**, and (d) **PdAu8-Mo6**, and simulated UV-Vis-NIR spectra of (e) **PdAu8-C** and (f) **PdAu8-B** by TD-DFT calculation. **D** UV-Vis-NIR spectra of (a) **PtAu8-NO3** in ethanol solution, DR-UV-Vis-NIR spectra of (b) **PtAu8-NO3**, (c) **PtAu8-PMo12**, and (d) **PtAu8-Mo6**, and simulated UV-Vis-NIR spectra of (e) **PtAu8-C** and (f) **PtAu8-B**.

structural isomerization from crown-motif to butterfly-motif also occurs in the case of **PdAu8** and **PtAu8** by the association with **Mo6**, and an absorption peak appears at longer-wavelength (703 nm for **PdAu8-Mo6**, 600–700 nm for **PtAu8-Mo6**), as shown in Fig. 1.

**Stable structures and optical properties of PdAu8 and PtAu8.** The stable structures of **PtAu8-C** and **PtAu8-B** were calculated in this study (Fig. 2). **Au9-C, PdAu8-C, Au9-B,** and **PdAu8-B** were also optimized at the same calculation level (see Supplementary Fig. S6) although we had calculated them previously[32]. The optimized structure of **PtAu8-C** in Fig. 2 agrees with that determined by single-crystal XRD[34]. In addition, **PtAu8-B** can be generated as a stable entity like **PdAu8-B** as shown in Fig. 2. In all cases, the crown-motif structure was more stable than the butterfly-motif in Fig. 2 and Supplementary Fig. S6. Note that the energy difference between the crown-motif and butterfly-motif in **PtAu8** (0.30 eV) was greater than those in **Au9** (0.23 eV) and **PdAu8** (0.20 eV).

The cores of crown-motif and butterfly-motif **Au9, PdAu8,** and **PtAu8** are oblate superatoms with six valence electrons[32,36].

Supplementary Fig. S7 shows the energy diagram and superatomic orbitals for all clusters. The electronic states near the frontier orbitals of **Au9-C, PdAu8-C,** and **PtAu8-C** were very similar. The highest occupied molecular orbitals (HOMOs) of the crown-motif structure are composed of degenerate superatomic $1P_x$ and $1P_y$ orbitals and an unoccupied $1P_z$ orbital is located at the lowest unoccupied molecular orbital (LUMO) level. The superatomic 1D levels are shown at higher energy levels than LUMO. **Au9-C, PdAu8-C,** and **PtAu8-C** have the electronic configurations of $1S^21P^4$, which are consistent with those for oblate superatoms[36]. The isomerization from crown-motif to butterfly-motif induces the splitting of degenerated $1P_x$ and $1P_y$ in HOMO and $1D_{xy}$ and $1D_{x2-y2}$ in LUMO + 1, stabilization of $1D_{x2-y2}$, and destabilization of $1P_z$ in all clusters. As a result, LUMO in **Au9-B** and **PdAu8-B** becomes $1D_{x2-y2}$. Meanwhile, the LUMO in **PtAu8-B** is still a $1P_z$ superatomic orbital because the energy gap between LUMO ($1P_z$) and LUMO + 1 ($1D_{xy}$ and $1D_{x2-y2}$) in **PtAu8-C** is larger than those in **Au9-C** and **PdAu8-C**. These electronic changes in frontier orbitals by isomerization affect the optical properties. The optical absorptions of all clusters were calculated by time-dependent (TD)-DFT as shown in Fig. 1 and Supplementary Fig. S1. Previously, we reported that the

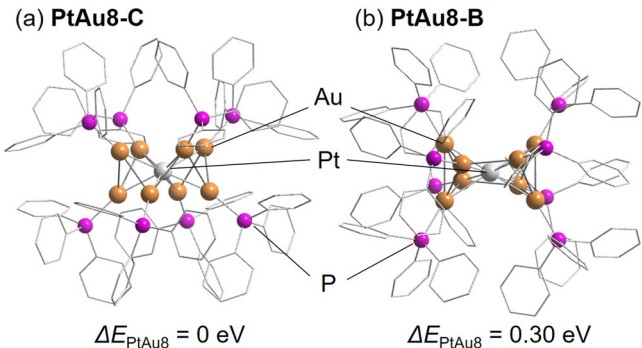

**Fig. 2 Optimized structures. a PtAu8-C** and **b PtAu8-B**. Color code: P (pink), Au (gold), and Pt (white). Phenyl rings are shown by gray wire frames. H atoms are omitted for easy to see. The relative energies with respect to the crown-motifs are shown.

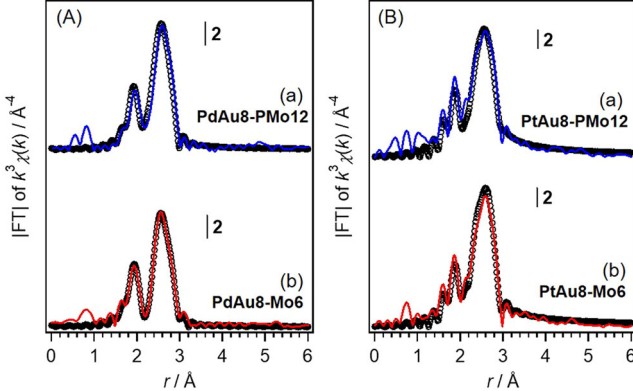

**Fig. 3 FT-EXAFS spectra. A** Au $L_3$-edge FT-EXAFS of (a) **PdAu8-PMo12** and (b) **PdAu8-Mo6** measured at 10 K. **B** Au $L_3$-edge FT-EXAFS of (a) **PtAu8-PMo12** and (b) **PtAu8-Mo6** measured at 10 K. The circles represent the fitting curves, whose parameters and results are listed in Table 1.

optical gap of **Au9-C** (Supplementary Fig. S1a, c) could be assigned to electron transition from HOMO($1P_x$, $1P_y$) to LUMO + 1($1D_{x2-y2}$) in Supplementary Fig. S1e and the characteristic peak at 703 nm observed in **Au9-B** (Supplementary Fig. S1b) is attributable to electron transition from HOMO($1P_y$) to LUMO + 1($1D_{x2-y2}$) at 687 nm in Fig. S1f[32]. The characteristic peaks in DR-UV-Vis-NIR spectra of **PdAu8-NO3/PdAu8-PMo12** with a crown-motif (530, 460, 417, 351 nm) and **PtAu8-NO3/PtAu8-PMo12** with crown-motif (482, 432, 404, 358 nm) in Fig. 1 were reproduced by TD-DFT calculations, as shown in Fig. 1Cf (456, 421, 394, 352 nm) and Fig. 1Df (475, 416, 386, 362 nm). Interestingly, the simulated optical absorptions of **PdAu8-B** and **PtAu8-B** have peaks at a longer wavelength (648 nm for **PdAu8-B** and 568 nm for **PtAu8-B**), assigned to electron transition from HOMO($1P_y$) to $1D_{x2-y2}$, than **PdAu8-C** and **PtAu8-C**. Therefore, the absorptions at 703 nm in **PdAu8-Mo6** (Fig. 1Cd) and at 600–700 nm in **PtAu8-Mo6** (Fig. 1Dd), which appeared by the formation of a composite of **PdAu8/PtAu8** with **Mo6**, are strongly suggested to be the optical gap of **PdAu8-B** and **PtAu8-B**.

**Local structures of PdAu8-Mo6 and PtAu8-Mo6.** The structural difference between crown-motif and butterfly-motif can be characterized by X-ray absorption fine structure (XAFS) because the coordination number (CN) of lateral Au–Au bonds (CN = 3) in the core of **PdAu8-B** and **PtAu8-B** is larger than that (CN = 2) in **PdAu8-C** and **PtAu8-C**, whereas the CNs of radial Au–Pd(Pt) bonds and Au–P bonds are 1.0 and 1.0, respectively from the Au site and the CN of radial Pd(Pt)–Au are 8.0 from Pd(Pt) site. Figure 3 shows the Au $L_3$-edge FT-EXAFS spectra of **PdAu8-PMo12, PdAu8-Mo6, PtAu8-PMo12,** and **PtAu8-Mo6** measured at 10 K, obtained from the EXAFS oscillations as shown in Supplementary Fig. S8. The Pd K-edge and Pt $L_3$-edge EXAFS and FT-EXAFS spectra are also shown in Supplementary Fig. S9. Table 1 and Supplementary Table S1 show the results of curve fitting analysis. First, Pd K- and Pt $L_3$-edges FT-EXAFS analysis revealed that the Pd in **PdAu8-PMo12** and **PdAu8-Mo6**, and Pt in **PtAu8-PMo12** and **PtAu8-Mo6** were located at the center of the core because the Pd–P and Pt–P bonds did not observe and the CNs of Pd–Au and Pt–Au were *ca* 8.0, as shown in Supplementary Fig. S9 and Supplementary Table S1. The CN and bond distance ($r$) for all bonds obtained by the curve fitting analysis of **PdAu8-PMo12** in Table 1 and Supplementary Table S1 agreed with those determined by single-crystal XRD as shown in Supplementary Table S2, and the CN of Au–Au in **PdAu8-PMo12** with a crown-motif was 2.0 ± 0.2. A similar result was obtained

for **PtAu8-PMo12** (CN of Au–Au: 2.0 ± 0.2). Meanwhile, the CNs of Au–Au in **PdAu8-Mo6** and **PtAu8-Mo6** were 3.1 ± 0.2 and 2.9 ± 0.2, respectively, which are characteristic CNs of Au–Au (CN = 3) for butterfly-motif **PdAu8-B** and **PtAu8-B** in Supplementary Table S2. We also analyzed the local structures of **Au9-NO3, Au9-PMo12, Au9-Mo6, PdAu8-NO3,** and **PtAu8-NO3** by XAFS. The Au $L_3$-, Pd-, and Pt $L_3$-edges EXAFS and FT-EXAFS spectra are shown in Supplementary Figs. S8–S10. The results of curve fitting analysis for **Au9-NO3, Au9-PMo12, Au9-Mo6, PdAu8-NO3,** and **PtAu8-NO3** are listed in Supplementary Tables S1 and S3. The structural parameters indicate that **Au9-NO3** has a butterfly-motif structure and the others have a crown-motif one, which is in agreement with the structures determined by single-crystal XRD analysis[31,34,37].

## Discussion

**Structures of PdAu8-Mo6 and PtAu8-Mo6.** It is known that crown-motif **Au9-C** can be isomerized to butterfly-motif **Au9-B** by the association of [NO3]− and Cl− anions[31,32]. In contrast, to the best of our knowledge, the structural isomers of **PdAu8-C** and **PtAu8-C** have not been reported to date because the formation of stiff M–Au (M = Pd, Pt) bonds by the hetero metal doping into **Au9-C** suppresses the isomerization[32]. In this study, we found that the composites of **PdAu8-Mo6** and **PtAu8-Mo6** showed different optical absorption properties from **PdAu8-PMo12** and **PtAu8-PMo12**, which had crown-motif structures (see Fig. 1)[34]. Under the same condition, **Au9-C** in **Au9-NO3** solution was isomerized to form butterfly-motif **Au9-Mo6** composite by the association with **Mo6**, as shown in Supplementary Fig. S1. DFT calculation revealed that both **PdAu8-B** and **PtAu8-B** structures are stable entities and the optical absorptions of **PdAu8-Mo6** and **PtAu8-Mo6** are explained by TD-DFT analysis of **PdAu8-B** and **PtAu8-B**, respectively. XAFS analysis suggested that the local structures of **PdAu8-Mo6** and **PtAu8-Mo6** were in good accordance with those of **PdAu8-B** and **PtAu8-B**, respectively. From the above results, we concluded that the butterfly-motif **PdAu8-B** and **PtAu8-B** were successfully synthesized by the isomerization of crown-motif **PdAu8-C** and **PtAu8-C** using **Mo6** as a counteranion.

**Discussion on structural isomerization in PdAu8 and PtAu8.** We previously reported that the structural isomerization from **PdAu8-C** to **PdAu8-B** is suppressed in **PdAu8** because the activation energy for structural isomerization in **PdAu8** becomes larger than that in **Au9** by the presence of stiff Pd–Au bonds[32].

**Table 1 Curve fitting results of Au L$_3$-edge FT-EXAFS for PdAu8-Mo6, PdAu8-PMo12, PtAu8-Mo6 and PtAu8-PMo12, and $\theta_E$ values for PdAu8-PMo12 and PtAu8-PMo12.**

| Sample | Bond | CN | r/Å | $\sigma^2$ | R-factor (%) | $\theta_E$ (K) |
|---|---|---|---|---|---|---|
| PdAu8-PMo12 | Au–P | 1.0 (2) | 2.29 (4) | 0.05 (4) | 4.2 | 390 (90) |
| | Au–Pd | 1.0 (1) | 2.63 (3) | 0.05 (2) | | 210 (18) |
| | Au–Au | 2.0 (2) | 2.80 (2) | 0.06 (2) | | 126 (11) |
| PdAu8-Mo6 | Au–P | 1.1 (2) | 2.30 (4) | 0.05 (4) | 2.5 | |
| | Au–Pd | 1.0 (2) | 2.63 (3) | 0.06 (3) | | |
| | Au–Au | 3.1 (2) | 2.80 (3) | 0.07 (2) | | |
| PtAu8-PMo12 | Au–P | 1.0 (2) | 2.20 (3) | 0.04 (4) | 12.8 | 376 (52) |
| | Au–Pt(Au) | 1.0 (1) | 2.66 (2) | 0.03 (2) | | 218 (36) |
| | Au–Au | 2.0 (2) | 2.79 (2) | 0.06 (2) | | 127 (14) |
| PtAu8-Mo6 | Au–P | 1.0 (2) | 2.20 (4) | 0.05 (4) | 10.9 | |
| | Au–Pt(Au) | 1.1 (1) | 2.65 (2) | 0.04 (2) | | |
| | Au–Au | 2.9 (2) | 2.81 (3) | 0.07 (2) | | |

*CN coordination number, r bond distance, $\sigma^2$ Debye–Waller factor.*
*Numbers in parentheses represent uncertainties. The reliability factor (R-factor) is defined as:*
*R-factor ={$\Sigma[k^3 \chi_{obs}(k) - k^3 \chi_{cal}(k)]^2 / \Sigma[k^3 \chi_{obs} (k)]^2\}^{1/2}$ where, $\chi_{obs}$ and $\chi_{cal}$ correspond to the observed and calculated data, respectively.*

Since **PtAu8-NO3** had a crown-motif structure like **PdAu8-NO3**, it was expected that radial Pt–Au bonds in **PtAu8** are also stiffer than radial Au–Au bonds in **Au9**. The Pt–Au bond stiffness in **PtAu8-PMo12** with a crown-motif structure was evaluated using the Einstein temperature ($\theta_E$), which was determined by the temperature dependence of Debye–Waller (DW) factors[32]. The $\theta_E$ for each bond was obtained from curve fitting analysis of Au L$_3$- and Pt L$_3$-edges FT-EXAFS measured at 10–300 K. Supplementary Fig. S11 shows the temperature dependence of DW factors for Au–P, Au–Pt, and Au–Au from Au L$_3$-edge, and Pt–Au from the Pt L$_3$-edge. The large temperature dependence of DW factors of Au–Pt and Au–Au bonds represented the thermal-induced fluctuation of **PtAu8-PMo12**. The $\theta_E$ values of Au–P, Au–Pt, Au–Au, and Pt–Au bonds are shown in Table 1 and Supplementary Table S1. In the same way, we evaluated the $\theta_E$ values of Au–P, Au–Pd, Au–Au, and Pd–Au bonds in **PdAu8-PMo12** with a crown-motif structure using the temperature dependence of DW factors (Supplementary Fig. S11), as shown in Table 1 and Supplementary Table S1. The obtained $\theta_E$ values in **PdAu8-PMo12** were comparable to those in crown-motif [PdAu$_8$(PPh$_3$)$_8$]Cl$_2$ reported previously[32]. The $\theta_E$ values (218 K from the Au L$_3$-edge, 207 K from the Pt L$_3$-edge) of Au–Pt in crown-motif **PtAu8-PMo12** were similar to those (210 K from the Au L$_3$-edge and 212 K from the Pd K-edge) of Au–Pd in a crown-motif **PdAu8-PMo12**. Since the $\theta_E$ of radial Au–Au in **Au9-PMo12** with a crown-motif was reported to be 181 K[32], the radial M–Au (M = Pd, and Pt) bonds in **PdAu8-C** and **PtAu8-C** were stiffer than radial Au–Au in **Au9-C**. In addition, the energy difference between the crown-motif and the butterfly-motif in **PtAu8** was largest among all clusters, as shown in Fig. 2 and Supplementary Fig. S6. From these results, the activation energy of structural isomerization is predicted, as shown in Fig. 4. The stiffness of the bond is related to the steepness of the potential curve along the reaction coordinate. Thus, the activation energy is estimated to be in the order of **PtAu8** > **PdAu8** > **Au9**. It was assumed that the electrostatic interaction of cationic metal clusters with compact anions such as Cl$^-$ and [NO$_3$]$^-$ induces the structural isomerization from butterfly-motif to crown-motif in **Au9**[32], but the structural isomerization of **PdAu8** is suppressed by the stiffer Pd–Au bonds, which enhances the activation energy shown in Fig. 4. Therefore, **PdAu8-NO3** and **PtAu8-NO3** had a crown-motif structure. In this study, we demonstrated that **Mo6** could induce the structural isomerization from crown-motif to butterfly-motif for not only **Au9** but also **PdAu8** and **PtAu8**, which have stiff M–Au bonds in the cluster core.

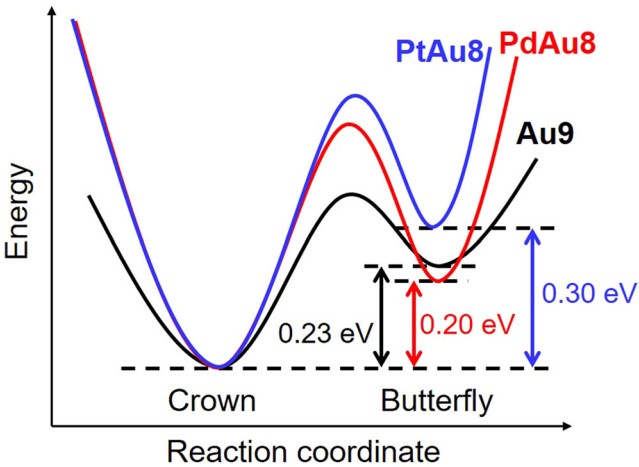

**Fig. 4 Potential curves of isomerization reaction.** Schematic image of potential curves of isomerization reaction for **Au9**, **PdAu8**, and **PtAu8**.

This isomerization was not explained by the electrostatic interaction of metal clusters with **Mo6**. Schulz-Dobrick and Jansen reported that butterfly-motif **Au9-B** was synthesized using keggin-type [PW$_{12}$O$_{40}$]$^{3-}$ by controlling the packing of clusters in the crystal[37]. The rock salt-type structure (six anions are coordinated to **Au9-B** octahedrally, see Supplementary Fig. S12a) is a key to the isomerization of **Au9-C** to **Au9-B** because **Au9-B** is oriented in such a way that the eight PPh$_3$ ligands, which surround the Au$_9$ core in an almost cubic fashion, point towards the eight faces to the coordination octahedron with [PW$_{12}$O$_{40}$]$^{3-}$. In fact, **Au9-PMo12**, **PdAu8-PMo12**, and **PtAu8-PMo12**, which have cesium chloride-type structure (eight **PMo12** are coordinated to cluster, see Supplementary Fig. S12b) had crown-motif structures[34]. The diameter of butterfly-motif **PdAu8-B**, and **PtAu8-B** was estimated to be *ca.* 1.7 nm (1.60–1.80 nm). If the clusters have the cubic closed packing (ccp) geometry, octahedral site (space) in ccp has 0.66–0.74 nm diameter. Since the diameter of **Mo6** is *ca.* 0.80 nm which is close to the size of octahedral site, **Mo6** can occupy the octahedral site in the ccp of **Au9-B**, **PdAu8-B**, and **PtAu8-B**. Single-crystal X-ray diffraction analysis revealed that **PdAu8-Mo6** had a rock salt-type structure, as shown in Supplementary Fig. S13, although the PPh$_3$ ligands and **Mo6** were disordered. The power X-ray diffraction patterns also demonstrated that **Au9-Mo6** and **PtAu8-Mo6** had same rock salt-type

structures as **PdAu8-Mo6** (Supplementary Fig. S14). Interestingly, the composites of [PdAu$_8$(PPh$_3$)$_8$]$^{2+}$ (**PdAu8-W6**), [PtAu$_8$(PPh$_3$)$_8$]$^{2+}$ (**PtAu8-W6**), and [Au$_9$(PPh$_3$)$_8$]$^{3+}$ (**Au9-W6**) with [W$_6$O$_{19}$]$^{2-}$, which has as same Lindqvist structure as **Mo6**, also had a butterfly-motif structure with rock salt-type structure because DR-UV-Vis-NIR and XRD patterns of **PdAu8-W6**, **PtAu8-W6**, and **Au9-W6** were similar to those of **PdAu8-Mo6**, **PtAu8-Mo6**, and **Au9-Mo6**, as shown in Supplementary Fig. S15. In addition, the **PdAu8-B** (**PtAu8-B**) in **PdAu8-Mo6** (**PtAu8-Mo6**) was reversibly isomerized to **PdAu8-C** (**PtAu8-C**) by the dissolution into DMSO, as shown in Supplementary Fig. S16. Therefore, the reason why the isomerization of **PdAu8-C** (**PtAu8-C**) to **PdAu8-B** (**PtAu8-B**) was induced by association with **Mo6** is that **PdAu8-B** (**PtAu8-B**) becomes more stable by the formation of a rock salt-type structure, which adds strong steric stress to the clusters to overcome the large activation energy for isomerization.

## Conclusions

In this study, the composites between **PdAu8/PtAu8** cations and **Mo6** anions have been synthesized to control the structures of **PdAu8/PtAu8**. It was found that the optical properties of **PdAu8-Mo6** and **PtAu8-Mo6** composites differed from those of crown-motif structures, respectively. DFT calculations and XAFS analysis revealed that the **PdAu8-C** and **PtAu8-C** in the solution were isomerized to butterfly-motif structures by the association with **Mo6** although it was predicted that the structural isomerization from crown-motif to butterfly-motif structure was suppressed in **PdAu8** and **PtAu8** by the higher activation energy and stiffer radial M–Au (M = Pd, Pt) than **Au9**. Single-crystal and powder X-ray diffraction analyses suggested that the **PdAu8-Mo6** and **PtAu8-Mo6** had a rock salt-type packing, which differed from crown-motif **PdAu8-PMo12** and **PtAu8-PMo12** having a cesium chloride-type packing (eight **Mo6** are coordinated to **PdAu8** and **PtAu8**). Six coordination of **Mo6** to **PdAu8** and **PtAu8** stabilizes the semi-stable butterfly-motif structure to overcome high activation energy for structural isomerization.

## Methods

**Chemicals**. All reagents were used as received, without further purification. Hydrogen tetrachloroaurate(III) tetrahydrate (HAuCl$_4$·4H$_2$O, 99.0%) was purchased from Kanto Chemical Co., Inc. Silver(I) nitrate (AgNO$_3$, >99.9%) was purchased from Kojima Chemicals Co., Ltd. Tetrabutyl ammonium bromide {TBABr, [N(C$_4$H$_9$)$_4$]Br, >98.0%}, triethylamine [N(C$_2$H$_5$)$_3$, >99.0%], tetrakis(triphenylphosphine)palladium(0) [Pd(PPh$_3$)$_4$, >97.0%], and tetrakis(triphenylphosphine)platinum(0) [Pt(PPh$_3$)$_4$, >97.0%] were purchased from Tokyo Chemical Industry Co., Ltd. Sodium borohydride (NaBH$_4$, 95.0%), triphenylphosphine (PPh$_3$, 97.0%), 12 molybdo(VI) phosphoric acid n-hydrate [H$_3$(PMo$_{12}$O$_{40}$)·nH$_2$O, >95.0%], disodium molybdate(VI) dihydrate (Na$_2$MoO$_4$·2H$_2$O, >99.0%), disodium tungstate(VI) dihydrate (Na$_2$WO$_4$·2H$_2$O, 99.0%–100.5%), hydrochloric acid (HCl, 35.0%–37.0%), and acetic anhydride [(CH$_3$CO)$_2$O, >97.0%] were purchased from Wako Pure Chemical Industry.

**Synthesis of Au9/MAu8-NO3 (M = Pd, Pt) clusters**. [Au$_9$(PPh$_3$)$_8$](NO$_3$) (**Au9-NO3**) was synthesized by our previously reported procedure[32]. NaBH$_4$ (0.2 mmol) ethanolic solution (14 mL) was added dropwise to the suspension of AuPPh$_3$NO$_3$ (0.6 mmol) in ethanol (24 mL). After stirring for 2 h at room temperature, the solution was filtrated by a membrane filter (pore diameter = 0.20 μm). The filtrate was evaporated, and the precipitate was dissolved in dichloromethane (5 mL). After filtration and evaporation, the precipitate was washed with tetrahydrofuran and hexane. The green powder was obtained by vacuum drying.

[PdAu$_8$(PPh$_3$)$_8$](NO$_3$)$_2$ (**PdAu8-NO3**) was synthesized by previously reported procedures[34,38]. Solid-form NaBH$_4$ (0.4 mmol) was slowly added into the suspension of AuPPh$_3$NO$_3$ (0.3 mmol) and Pd(PPh$_3$)$_4$ (0.1 mmol) in ethanol (12 mL). After stirring for 1 h at room temperature, the solution was added to hexane (200 mL). The precipitated brown solid was washed with hexane and, then with pure water at least three times. After extraction of the residue with ethanol, a dark brown solid was obtained by evaporation of the solvent.

[PtAu$_8$(PPh$_3$)$_8$](NO$_3$)$_2$ (**PtAu8-NO3**) was synthesized by a slightly modified version of a previously reported procedure[39]. AuPPh$_3$NO$_3$ (1.0 mmol) and Pt(PPh$_3$)$_3$ (0.2 mmol) were added to tetrahydrofuran (20 mL) and stirred for 2 h

under bubbling H$_2$ gas at room temperature. The precipitate was collected by centrifugation (2500 rpm, 3 min). The obtained solid precipitate was recrystallized from dichloromethane and diethyl ether three times. Vacuum drying of red-orange precipitate gave a mixture of [HPtAu$_7$(PPh$_3$)$_8$](NO$_3$)$_2$ and [PtAu$_8$(PPh$_3$)$_8$](NO$_3$)$_2$. This mixture (169 mg) and AuPPh$_3$NO$_3$ (0.1 mmol) were dissolved in dichloromethane (10 mL), and N(C$_2$H$_5$)$_3$ (0.2 mmol) was added into this solution. After stirring for 24 h at room temperature, the solution was evaporated. The residual solid was recrystallized with diethyl ether three times. Finally, brown solid was obtained after vacuum drying.

**Synthesis of polyoxometalates**. (TBA)$_2$[Mo$_6$O$_{19}$] (**TBA-Mo6**) was prepared by a slightly modified version of a previously reported process[40,41]. Na$_2$MoO$_4$·2H$_2$O (10.3 mmol) in pure water (10 mL) was acidified with 6 M HCl (2.9 mL) at room temperature. In addition to TBABr (4.0 mmol) aqueous solution (2 mL), white precipitate immediately formed. After stirring for 1 h at 348 K, a yellow precipitate formed and was collected by centrifugation (3500 rpm, 5 min) and washed with water and methanol three times each. The product was dissolved in acetone and recrystallized by freezing at 213 K.

(TBA)$_3$[PMo$_{12}$O$_{40}$] (**TBA-PMo12**) was fabricated by cation exchange using the reported procedure with slight modification[37,40,42,43]. Solid TBABr (1.2 mmol) was added to H$_3$[PMo$_{12}$O$_{40}$]·nH$_2$O (0.3 mmol) aqueous solution (50 mL). After stirring for 0.5 h at room temperature, the precipitate was collected by centrifugation and washed with pure water. The precipitate was recrystallized from acetone (5 mL) and hexane (45 mL). Yellow powder was obtained by vacuum drying.

(TBA)$_2$[W$_6$O$_{19}$] (**TBA-W6**) was also prepared according to a reported procedure with a slight modification[40,44]. A mixture of Na$_2$WO$_4$·2H$_2$O (10.0 mmol) and acetic anhydride (4 mL) in N,N-dimethylformamide (DMF, 3 mL) was stirred at 373 K for 3 h. Then, a mixed acid of acetic anhydride (2 mL) and 12 M HCl (1.8 mL) in DMF (5 mL) was slowly added dropwise to the dispersion. The undissolved solid was removed by centrifugation and was washed with methanol (5 mL). After cooling the filtrate to room temperature, TBABr (5.2 mmol) methanolic solution was added dropwise with rapid stirring. The resulting precipitate was centrifugated and washed with methanol and diethyl ether. The white solid was obtained by vacuum drying.

**Synthesis of Au9/MAu8-POM (M = Pd, Pt; POM = Mo6, PMo12) composites**. **Au9/MAu8-POM** composites were synthesized by slightly modified versions of procedures reported in the literature[37,43]. Acetone solutions (20 mL) of **Mo6**, **PMo12** and **W6** (6 μmol each) were each mixed with acetone solutions (10 ml) of **Au9/MAu8-NO3** (4 μmol each). The precipitate was collected by centrifugation (2500 rpm, 3 min) and then washed with acetone. After drying, nine types of gold cluster salts, **Au9/MAu8-Mo6, Au9/MAu8-PMo12** (M = Pd, Pt), and **Au9/MAu8-W6**, were obtained.

**Characterizations**. The fabrication of gold clusters and polyoxometalates, and their composites was confirmed using the following techniques. The UV-Vis-NIR spectra of **Au9/MAu8-NO3** in ethanol were recorded in transmittance mode (V-770; Jasco). The UV-Vis-NIR spectra of **Au9/MAu8-Mo6, Au9/MAu8-PMo12**, and **Au9/MAu8-Mo6** were measured in diffuse reflectance (DR) mode. Electrospray ionization (ESI) mass spectra of **Au9/MAu8-NO3** in acetonitrile were measured in positive-ion mode using a time-of-flight (TOF) mass spectrometer (micrOTOF-II; Bruker). The ligation of PPh$_3$ to gold clusters was confirmed using a $^{31}$P NMR spectrometer (AV500; Bruker). The crystal structures of gold clusters were determined using a powder X-ray diffractometer (MiniFlex600; Rigaku). Crystal structure of **PdAu8-Mo6** was analyzed by a single-crystal x-ray diffractometer (sc-XRD) equipped at the BL40XU beamline in SPring-8. The crystal packing was solved by a direct method and refined by the full-matrix least-squares method using the Yadokari-XG crystallographic software[45]. The thermal properties of **TBA-Mo6** and **TBA-PMo12** were measured using thermal gravity-differential thermal analyzer (TG-DTA, STA 2500 Regulus; Netzsch).

The X-ray absorption fine structure (XAFS) spectra of **Au9/MAu8-Mo6** and **Au9/MAu8-PMo12** were measured at the BL01B1 beamline of the SPring-8 facility of the Japan Synchrotron Radiation Research Institute, where Si(111) double-crystal monochrometers were used to obtain the incident X-ray beam for Au L$_3$-edge XAFS measurements. All samples were pressed into a pellet and mounted on a copper holder attached to the cryostat. Au L$_3$-edge XAFS spectra were measured in the transmission mode using ionization chambers at 10–300 K. In the case of Pd K- and Pt L$_3$-edges XAFS measurements, spectra were collected in fluorescence mode using ionization chamber and 19-element Ge solid state detector. The data reduction was conducted using the xTunes software[46] for extended X-ray absorption fine structure (EXAFS), Fourier transform (FT)-EXAFS, and curve fitting analyses and the REX2000 Ver. 2.5.9 program (Rigaku Co.) for the analysis of temperature-dependence of DW factors. The $k^3$-weighted χ spectra in the $k$ range of 3.0–18.5 Å$^{-1}$ for Au L$_3$-edge, 3.0–17.0 Å$^{-1}$ for Pd K-edge, and 3.0–9.0 Å$^{-1}$ for Pt L$_3$-edge were Fourier transformed into $r$ space to obtain FT-EXAFS spectra. The curve fitting analysis was conducted in the $r$ range of 1.7–2.9 Å for Au L$_3$-edge, 2.2–2.8 Å for Pd K-edge, and 1.7–3.2 Å for Pt L$_3$-edge. **Au9-C, Au9-B, MAu8-C**, and **MAu8-B** have Au-P, radial (short) Au–Au (**Au9-C, Au9-B**) and Au-M (**MAu8-C, MAu8-B**), and lateral (long) Au–Au bonds in the clusters, as shown in

Supplementary Table S2. Thus, the phase shifts and backscattering amplitude functions for Au–P, Au–Au, Au–Pd, Au–Pt, Pd–Au, and Pt–Au were extracted from $Au_2P_3$ (ICSD#8058), Au metal (ICSD#44362), and $PdAu_{24}(SCH_3)_{18}$[47] and $PtAu_{24}(SCH_3)_{18}$, whose structure was constracted by the substitution of Pd with Pt in $PtAu_{24}(SCH_3)_{18}$, using the FEFF8.5 L program[48]. The Au $L_3$-edge FT-EXAFS data were fitted using the calculated phase shifts and backscattering amplitude functions of Au–P, short Au–M (M = Au, Pd, Pt), and long Au–Au bonds according to the previous report[32] and model structures, whose structual parameters are shown in Supplementary Table S2. In the case of Pd K- and Pt $L_3$-edges FT-EXAFS, Pd–Au and Pt–Au bonds were applied for curve fitting analysis, respectively because the Pd–P and Pt–P bonds did not appear in the Pd K- and Pt $L_3$-edges FT-EXAFS spectra. Amplitude reduction factor, $S_0$, of 1.0 was used for the curve fitting analysis. The DW values were evaluated from the FT-EXAFS at each temperature obtained from EXAFS data ($3.0 \leq k \leq 16.0$ for Au $L_3$- and Pd K-edges, $3.0 \leq k \leq 9.0$ for Pt $L_3$-edge) according to the previous work[32]. The values of $r$ and DW at each temperature were determined by least-squares fit analysis while keeping the CN values the same as those obtained from the curve fitting analysis at 10 K[49] using the analytical EXAFS range of 3.0–18.5 Å$^{-1}$ for the Au $L_3$-edge, 3.0–17.0 Å$^{-1}$ for the Pd K-edge, and 3.0–9.0 Å$^{-1}$ for the Pt $L_3$-edge, respectively.

The Debye–Waller factor ($\sigma^2$) consists of static ($\sigma_S^2$) and dynamic ($\sigma_D^2$) components, which arise from temperature-independent structural disorder and temperature-dependent atomic oscillation, respectively[31]. According to the Einstein model that assumes three independent harmonic oscillator with different Einstein temperatures ($\theta_E$) for Au–M (M = central Au, Pd, and Pt), Au–Au, and Au–P bonds. $\sigma^2$ is expressed as follows:

$$\sigma^2 = \sigma_S^2 + \sigma_D^2$$

$$\sigma^2 = \sigma_S^2 + \frac{h^2}{8\pi^2 \mu k_B \theta_E} \coth \frac{\theta_E}{2T}$$

where $h$, $k_B$, $\mu$, and $T$ represent the Planck constant, Boltzmann constant, reduced mass of adjacent atoms, and temperature, respectively. The $\theta_E$ values were determined by fitting the temperature dependence of the DW factors for each bond.

Density functional theory (DFT) calculations were conducted using the Gaussian 09 program[50]. Electronic and geometric structures of $[Au_9(PPh_3)_8]^{3+}$ (Au9), $[MAu_8(PPh_3)_8]^{2+}$ (MAu8, M = Pd, Pt), $NO_3^-$, $[Mo_6O_{19}]^{2-}$ (Mo6), and $[PMo_{12}O_{40}]^{3-}$ (PMo12), were calculated using the B3LYP function. LanL2DZ for Au, Pd, Pt, and Mo atoms; 6–31 G* for C, H, and P (in Au9/MAu8); and 6–31 + G* for O and P (in PMo12) were used as basis sets. Structural optimization and frequency analyses were performed for crown-motif Au9/MAu8-C and the corresponding butterfly-motif. Optical properties of MAu8-B and MAu8-C were calculated by the time-dependent (TD) DFT method solving 40 singlet states.

## Data availability

All data generated or analyzed during this study are included in this article (and its Supplementary Information files).

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

## Acknowledgements

This study was partially supported by the Program for JSPS KAKENHI (No. 19KK0139, 21H01718, and 22K14543), TMU research fund for young scientists, and Tokyo Metropolitan Government Advanced Research (R3-1). The XAFS and SC-XRD measurements were performed at BL01B1 and BL40XU in SPring-8 under the approval of Synchrotron Radiation Research Institute (JASRI) (Nos. 2019B0948, 2020A0715, 2022A1532, 2022A1627, 2022B1684, and 2022B1911).

## Author contributions

S.Y. guided the whole experiment and conceived the idea. S.K. drafted the manuscript, Y.F. and T.M. conducted most of the experiments, H.T. analyzed SC-XRD, N.Y. and K.N. measured SC-XRD and XAFS respectively, and N.N. carried out DFT calculations. J.H. and Y.N. gave some advice for the characterization and analysis, and all the authors contributed to the final polishing of the manuscript.

## Competing interests

The authors declare no competing interests.
