## [Peer Review File · Communications Chemistry]

Reviewers' comments:

Reviewer #1 (Remarks to the Author):

Seiji Yamazoe and colleagues have found the butterfly structures of PdAu₈ and PtAu₈ by the structural isomerization from the crown analogues by using an anionic polyoxometalate. The study has been done well. The explanation of this phenomenon is logical. I will be happy to support the publication of this work.

I have only one minor comment to make. The paper appears to be silent about the reversibility of the transition in solution and solid states. An explicit mention of this will be useful.

The revised file need not be reviewed.

Reviewer #2 (Remarks to the Author):

In this study, Seiji Yamazoe and his co-workers did a very interesting work, they prepared the butterfly-motif MAu₈ (M= Au, Pt, Pd) clusters, which have never been reported yet. And a series of powerful proofs (UV-vis-NIR, ESI-TOF-MS, XRD, XANFS, DFT and so on) were present and discussed to support their points. This work is well organized and prepared. I think this work set a very good example for gold clusters research and would draw a great deal of attention from readers in this field. Therefore, I recommend publishing this work on the well-known journal of ****. But I have two questions.

1. As reported the reversible conversion between butterfly and Crown isomers of Au₉ cluster can be achieved with the help of Counter anions, will it go for the MAu₈ clusters?
2. You mentioned in manuscript that "The formation of stiff M–Au (M = Pd, Pt) bonds by the hetero metal doping into Au₉-C suppresses the isomerization", will the counter ions neutralize the effect of "stiff M–Au (M = Pd, Pt) bonds"? Is it possible to detect the interaction between cluster and counter ions in crystal structures? The related work on Au₉ should be cited, 10.1038/s42004-023-00817-5.

Reviewer #3 (Remarks to the Author):

Here the authors report an approach for controlling the structural isomerization of PdAu₈ and PtAu₈ clusters by forming composites between the cluster cations and Mo₆ anions. XAFS and DFT were used to determine the structural motif of the new isomers, and based on these techniques, the authors believe that the evidence suggests a crown to butterfly isomerization. This work, as it pertains to the control of nanocluster geometry, is extremely interesting, broadly important, and timely. Unfortunately, I cannot recommend this paper for publication as is, but it will be publishable after the XAFS fitting approach is clarified. The following comments are for the improvement of this work.

1. In the current state of the main text and SI, there is not enough information about the author's EXAFS fitting approach to access their interpretation of the EXAFS data. Details regarding the fitting model, such as how the amplitude reduction factor was selected, which variables were varied or held constant, what data was included in the fitting model (did you fit both edges simultaneously?), the k-range and r-range of fitting (did it change for each temperature?), and the FEFF simulation parameters (and a justification for why the parameters were selected), are important for giving the reader a full and transparent idea of the logic of interpretation. The interpretation of EXAFS data requires significant assumptions embedded in the fitting model. So it is also suggested that the authors justify these assumptions and acknowledge any limitations in their fitting model.

2. XAFS is an ensemble average technique, i.e., all the absorbing atoms in the sample are probed. The authors should comment somewhere in the main text on the homogeneity of the samples. The authors assume that Pd or Pt is in the center of the clusters, but it is unclear how justified that assumption is based on the current text. The authors should justify this assumption with preferably experimental or, at least, theoretical evidence that Pd/Pt exists at the center of the cluster, as this assumption is key to the interpretation of the author's XAFS data.

3. The authors analyze PtAu from the Au L3 edge, but do not mention in methods or SI how they handle the superposition of the Pt L3 edge over the Au L3 edge (sometimes referred to as "leakage" of the Pt signal into the Au signal). If this is not corrected for, the analysis of the Au L3 edge data will not be interpretable quantitatively (here is one way to correct for it: PHYSICAL REVIEW B 80, 064111 2009). Moreover, the analysis of the PtAu data seems to rely on the assumption that the Pt atom is in the center of the cluster. If this were not the case, the authors would not be able to separate the Pt-Pt and Pt-Au or Au-Au and Au-Au contributions from each other because Pt and Au are next to each other in the periodic tables and, thus, have very similar photoelectron backscattering properties. It is critical in this case that the authors justify the assumption that Pt is in the center of the cluster—perhaps via DFT. Otherwise, the Pt L3 and Au L3 edge data should be interpreted as Pt-M and Au-M contributions where M is Au and/or Pt.

4. The authors present the EXAFS fitting results for the samples at 10 K that match the butterfly geometry (the Au-Pd and Au-Au coordination numbers). The authors also report the very nice σ^2 vs. temperature plot in S11. Is there any reason not to report the coordination numbers for these temperatures? Do the coordination numbers remain the same with temperature? Only as a suggestion: commenting on the stability of this butterfly motif at experimentally-relevant conditions may improve the impact of the results.

The revised parts of our manuscript to response to the reviewers are highlighted in red in “the Supporting Information for Review Only”.

Response to the reviewer 1 comment:

Seiji Yamazoe and colleagues have found the butterfly structures of PdAu₈ and PtAu₈ by the structural isomerization from the crown analogs by using an anionic polyoxometalate. The study has been done well. The explanation of this phenomenon is logical. I will be happy to support the publication of this work.

We appreciate reviewer 1 for the positive comments. We revised our paper according to the comment. The following is the response for the comment.

(Comment 1) I have only one minor comment to make. The paper appears to be silent about the reversibility of the transition in solution and solid states. An explicit mention of this will be useful.

Reply: Thank you for your valuable comment. The butterfly-motif **PdAu₈-Mo₆** and **PtAu₈-Mo₆** solids were re-solved in DMSO or DMF and crown-motif **PdAu₈** and **PtAu₈** were re-formed in the solution. The UV-vis-NIR results of **PdAu₈-Mo₆** and **PtAu₈-Mo₆** solution in DMSO were added in Fig. S16. We added the point that the reversibility of the isomerization in the manuscript in section 3.2 as follows,

In addition, the **PdAu₈-B (PtAu₈-B)** in **PdAu₈-Mo₆ (PtAu₈-Mo₆)** was reversibly isomerized to **PdAu₈-C (PtAu₈-C)** by the dissolution into DMSO, as shown in Fig. S16.

Response to the reviewer 2 comments:

In this study, Seiji Yamazoe and his co-workers did a very interesting work, they prepared the butterfly-motif MAu₈ (M= Au, Pt, Pd) clusters, which have never been reported yet. And a series of powerful proofs (UV-vis-NIR, ESI-TOF-MS, XRD, XANFS, DFT and so on) were present and discussed to support their points. This work is well organized and prepared. I think this work set a very good example for gold clusters research and would drown a great deal of attention from readers in this field. Therefore, I recommend publishing this work on the well-known journal of ****. But I have two questions.

We would like to thank reviewer 2 for the positive comments. Manuscript was revised according to the comments. The followings are the responses for each comment one by one.

(Comment 1) As reported the reversible conversion between butterfly and Crown isomers of Au₉ cluster can be achieved with the help of Counter anions, will it go for the MAu₈ clusters?

Reply: Thank you for the comment. As you know, reversible isomerization between crown-motif and butterfly-motif was reported for Au₉ clusters. We confirmed that the butterfly-motif **PdAu₈-Mo₆** and **PtAu₈-Mo₆** solids were re-solved in DMSO or DMF and crown-motif **PdAu₈** and **PtAu₈** were re-formed in the solution. The UV-vis-NIR results of **PdAu₈-Mo₆** and **PtAu₈-Mo₆** solution in DMSO were added in Fig. S16. We added the point that the reversibility of the isomerization in the manuscript in section 3.2 as follows,

In addition, the **PdAu₈-B (PtAu₈-B)** in **PdAu₈-Mo₆ (PtAu₈-Mo₆)** was reversibly isomerized to **PdAu₈-C (PtAu₈-C)** by the dissolution into DMSO, as shown in Fig. S16.

(Comment 2) You mentioned in manuscript that “The formation of stiff M–Au (M = Pd, Pt) bonds by the hetero metal doping into Au₉-C suppresses the isomerization”, will the counter ions neutralize the effect of “stiff M–Au (M = Pd, Pt) bonds”? Is it possible to detect the interaction between cluster and counter ions in crystal structures? The related work on Au₉ should be cited, 10.1038/s42004-023-00817-5.

Reply: Thank you for the valuable comment. Now, we think that the bond stiffness of M–Au in MAu₈ clusters affects the activation energy of structural isomerization as shown in Fig. 4. So, the formation of stiff bonds increases the activation energy. The sc-XRD, powder XRD, XAFS analysis, and DFT calculations revealed that the packing (arrangement) of the clusters (MAu₈ cation and Mo₆ anion) in the solid is important and rock salt-type structure induces stronger energy to MAu₈ clusters than the activation energy of structural isomerization from crown-motif to butterfly-motif and stabilizes the butterfly-motif structure. The details are discussed in section 3.2. Please refer to section 3.2.

Response to the reviewer 3 comments:

Here the authors report an approach for controlling the structural isomerization of PdAu₈ and PtAu₈ clusters by forming composites between the cluster cations and Mo₆ anions. XAFS and DFT were used to determine the structural motif of the new isomers, and based on these techniques, the authors believe that the evidence suggests a crown to butterfly isomerization. This work, as it pertains to the control of nanocluster geometry, is extremely interesting, broadly important, and timely. Unfortunately, I cannot recommend this paper for publication as is, but it will be publishable after the XAFS fitting approach is clarified. The following comments are for the improvement of this work.

We would like to thank reviewer 3 for the valuable comments and suggestions. We revised our paper according to the comments. The followings are the responses for each comment one by one.

(Comment 1) In the current state of the main text and SI, there is not enough information about the author's EXAFS fitting approach to access their interpretation of the EXAFS data. Details regarding the fitting model, such as how the amplitude reduction factor was selected, which variables were varied or held constant, what data was included in the fitting model (did you fit both edges simultaneously?), the k-range and r-range of fitting (did it change for each temperature?), and the FEFF simulation parameters (and a justification for why the parameters were selected), are important for giving the reader a full and transparent idea of the logic of interpretation. The interpretation of EXAFS data requires significant assumptions embedded in the fitting model. So it is also suggested that the authors justify these assumptions and acknowledge any limitations in their fitting model.

Reply: Thank you very much for the significant point and comment. As the reviewer pointed out, the information on the curve fitting analysis has been insufficient. According to the reviewer's comment, we described the detailed procedure (range of curve fitting, the way how to calculate the backscattering amplitude and phase shift, model structures for curve fitting, how to evaluate the temperature-dependency of DW) of curve fitting analysis in the experimental sections. Following are the additional explanation for the XAFS analysis in experimental section,

In the case of Pd K- and Pt L₃-edges XAFS measurements, spectra were collected in fluorescence mode using ionization chamber and 19-element Ge solid state detector. The data reduction was conducted using the xTunes software⁴⁶ for extended X-ray absorption fine structure (EXAFS), Fourier transform (FT)-EXAFS, and curve fitting analyses and the REX2000 Ver. 2.5.9 program (Rigaku Co.) for the analysis of temperature-dependence of DW factors. The k^3 -weighted χ spectra in the k range of 3.0–18.5 Å⁻¹ for Au L₃-edge, 3.0–17.0 Å⁻¹ for Pd K-edge, and 3.0–9.0 Å⁻¹ for Pt L₃-edge were Fourier transformed into r space to obtain FT-EXAFS spectra. The curve fitting analysis was conducted in the r range of 1.7–2.9 Å for Au L₃-edge, 2.2–2.8 Å for Pd K-edge, and 1.7–3.2 Å for Pt L₃-edge. **Au9-C**, **Au9-B**, **MAu8-C**, and **MAu8-B** have Au–P, radial (short) Au–Au (**Au9-C**, **Au9-B**) and Au–M (**MAu8-C**, **MAu8-B**), and lateral (long) Au–Au bonds in the clusters, as shown in Table S2. Thus, the phase shifts and backscattering amplitude functions for Au–P, Au–Au, Au–Pd, Au–Pt, Pd–Au, and Pt–Au were extracted from Au₂P₃ (ICSD#8058), Au metal (ICSD#44362), and PdAu₂₄(SCH₃)₁₈⁴⁷ and PtAu₂₄(SCH₃)₁₈, whose structure was constructed by the substitution of Pd with Pt in PtAu₂₄(SCH₃)₁₈, using the FEFF8.5L program.⁴⁸ The Au L₃-edge FT-EXAFS data were fitted using the calculated phase shifts and backscattering amplitude functions of Au–P, short Au–M (M = Au, Pd, Pt), and long Au–Au bonds according to the previous report³² and model structures, whose structural parameters are shown in Table S2. In the case of Pd K- and Pt L₃-edges FT-EXAFS, Pd–Au and Pt–Au bonds were applied for curve fitting analysis, respectively because the Pd–P and Pt–P bonds did not appear in the Pd K- and Pt L₃-edges FT-EXAFS spectra. Amplitude reduction factor, S_0 , of 1.0 was used for the curve fitting analysis. The DW values were evaluated from the FT-EXAFS at each temperature obtained from EXAFS data ($3.0 \leq k \leq 16.0$ for Au L₃- and Pd K-edges, $3.0 \leq k \leq 9.0$ for Pt L₃-edge) according to the previous work³². The values of r and DW at each temperature were determined by least-squares fit analysis while keeping the CN values the same as those obtained from the curve fitting analysis at 10 K⁴⁹ using the analytical EXAFS range of 3.0–18.5 Å⁻¹ for the Au L₃-edge, 3.0–17.0 Å⁻¹ for the Pd K-edge, and 3.0–9.0 Å⁻¹ for the Pt L₃-edge respectively.

(Comment 2) XAFS is an ensemble average technique, i.e., all the absorbing atoms in the sample are probed. The authors should comment somewhere in the main text on the homogeneity of the samples. The authors assume that Pd or Pt is in the center of the

clusters, but it is unclear how justified that assumption is based on the current text. The authors should justify this assumption with preferably experimental or, at least, theoretical evidence that Pd/Pt exists at the center of the cluster, as this assumption is key to the interpretation of the author's XAFS data.

Reply: Thank you for your valuable comment. The purity of the samples is important to determine the cluster structures as the reviewer described. First, we investigated the purity of the samples by ESI-MS and NMR and we confirmed that each cluster had a single composition as shown in Fig. 1. The location of Pd and Pt in the MAu9 clusters were determined by Pd K- and Pt L₃-edges EXAFS analysis. The Pd-P/Pt-P bonds did not observe in Pd K- and Pt L₃-edges FT-EXAFS of all MAu9 samples. These results gave us the conclusion that the Pd and Pt are occupied at the center of the MAu9 clusters and are not located at the other positions. To emphasize the purity of the MAu9 clusters, the following sentence was added in the section 2.3,

First, Pd K- and Pt L₃-edges FT-EXAFS analysis revealed that the Pd for **PdAu8-PMo12** and **PdAu8-Mo6**, and Pt for **PtAu8-PMo12** and **PtAu8-Mo6** were located at the center of the core because the Pd-P and Pt-P bonds did not observe and CNs of Pd-Au and Pt-Au were *ca* 8.0, as shown in Fig. S9 and Table S1.

(Comment 3) The authors analyze PtAu from the Au L3 edge, but do not mention in methods or SI how they handle the superposition of the Pt L3 edge over the Au L3 edge (sometimes referred to as “leakage” of the Pt signal into the Au signal). If this is not corrected for, the analysis of the Au L3 edge data will not be interpretable quantitatively (here is one way to correct for it: PHYSICAL REVIEW B 80, 064111 2009). Moreover, the analysis of the PtAu data seems to rely on the assumption that the Pt atom is in the center of the cluster. If this were not the case, the authors would not be able to separate the Pt-Pt and Pt-Au or Au-Au and Au-Au contributions from each other because Pt and Au are next to each other in the periodic tables and, thus, have very similar photoelectron backscattering properties. It is critical in this case that the authors justify the assumption that Pt is in the center of the cluster—perhaps via DFT. Otherwise, the Pt L3 and Au L3 edge data should be interpreted as Pt-M and Au-M contributions where M is Au and/or Pt.

Reply: Thank you for this important comment. As the reviewer' point, it is difficult to distinguish the neighbor Pt and Au atoms in the curve fitting analysis. However, we confirmed that the PtAu₈ clusters contained single Pt atom by ESI-MS and NMR analysis as shown in Fig. 1. In addition, curve fitting analysis of Pt L₃-edge FT-EXAFS revealed that the single Pd/Pt is located at the center position in the clusters because of no Pt-P bonds. From these results, we determined that the Pt occupies only center position of the cluster core. The purity of the PtAu₈ clusters and location of Pt in the clusters were mentioned as shown in the reply of Comment 2.

(Comment 4) The authors present the EXAFS fitting results for the samples at 10 K that match the butterfly geometry (the Au-Pd and Au-Au coordination numbers). The authors also report the very nice σ^2 vs. temperature plot in S11. Is there any reason not to report the coordination numbers for these temperatures? Do the coordination numbers remain the same with temperature? Only as a suggestion: commenting on the stability of this butterfly motif at experimentally-relevant conditions may improve the impact of the results.

Reply: Thank you for your valuable suggestion. We added the analytical procedure of temperature-dependency of DW in the experimental section (please see the reply for comment 1). The DW value for each temperature was obtained by the use of fixed CN values which were determined by curve fitting analysis of FT-EXAFS measured at 10 K because the DW value is significantly affected by CN value as you know. In other words, the DW value for each bond at each temperature could be analyzed (obtained) by the use of fixed CN values because of the large thermal vibration of Au-Au and Au-M bonds. So, low temperature measurement is required for the ligand-protected gold clusters to analyze the local structures by XAFS reported previously [S. Yamazoe *et al.*, *Inorg. Chem.*, **56**, 8319–8325 (2017), S. Yamazoe *et al.*, *Nat. Commun.*, **7**, 10414 (2016)]. If the butterfly-motif of MAu₈ changes to other structures by measurement temperature, the DW values would diverge. Therefore, the butterfly-motif is kept even at room temperature. In fact, the optical property of the MAu₈-Mo₆ (M = Pd, Pt) measured at room temperature could be explained by DFT calculations using MAu₈-B structures.

REVIEWERS' COMMENTS:

Reviewer #2 (Remarks to the Author):

The questions have been well answered.

Reviewer #3 (Remarks to the Author):

The authors have fully addressed my comments. I can now recommend this paper for publication.